# Anti-stress Properties of Atypical Antipsychotics

**DOI:** 10.3390/ph13100322

**Published:** 2020-10-20

**Authors:** Alice Sanson, Marco A. Riva

**Affiliations:** Department of Pharmacological and Biomolecular Sciences, University of Milan, Via Giuseppe Balzaretti 9, 20133 Milan, Italy; alice.sanson@unimi.it

**Keywords:** schizophrenia, stress, atypical antipsychotics, neuroplasticity, HPA axis, monoamine

## Abstract

Stress exposure represents a major environmental risk factor for schizophrenia and other psychiatric disorders, as it plays a pivotal role in the etiology as well as in the manifestation of disease symptomatology. It may be inferred that pharmacological treatments must be able to modulate the behavioral, functional, and molecular alterations produced by stress exposure to achieve significant clinical outcomes. This review aims at examining existing clinical and preclinical evidence that supports the ability of atypical antipsychotic drugs (AAPDs) to modulate stress-related alterations. Indeed, while the pharmacodynamic differences between AAPDs have been extensively characterized, less is known on their ability to regulate downstream mechanisms that are critical for functional recovery and patient stabilization. We will discuss stress-related mechanisms, spanning from neuroendocrine function to inflammation and neuronal plasticity, which are relevant for the manifestation of schizophrenic symptomatology, and we will discuss if and how AAPDs may interfere with such mechanisms. Considering the impact of stress in everyday life, we believe that a better understanding of the potential effects of AAPDs on stress-related mechanisms may provide novel and important insights for improving therapeutic strategies aimed at promoting coping mechanisms and enhancing the quality of life of patients affected by psychiatric disorders.

## 1. Introduction: Stress and Mental Illness

The term “stress” indicates an experience that may be perceived as challenging, both emotionally and physiologically, which implies a whole-body reaction to such a challenging event [1,2]. The consequences of stress exposure depend on such “reaction”: indeed, the ability to respond or adapt to stress will allow “survival” under a broad meaning (eustress), whereas deleterious outcomes can be observed when the organism fails to adapt, particularly following prolonged and repeated exposure to stress (distress). Importantly, the inability to cope with stress may lead to increased susceptibility to psychiatric disorders [1,3].

Several aspects concur in the response to stress, such as genetic makeup, previous life experiences, and the timing of stress exposure, which modulate the ability of the subject to adapt and respond to the challenging situation. With regard to timing, it is known that early life stages, such as gestation, childhood, or adolescence, are very sensitive temporal windows for stress exposure, considering that such “experiences” may interfere with the developmental trajectories of selected brain regions and circuits leading to persistent changes that predispose the subject toward the development of psychiatric disorders [4,5,6,7]. Stress represents a potential threat also in adulthood since it may contribute to relapse in stabilized patients [8]. Indeed, there is a link between stress exposure and the exacerbation of schizophrenic and depressive symptoms [4,9,10], suggesting that mechanisms that are altered by stress have a close link with different psychopathologic domains, including mood and cognition.

It is therefore important to establish if and to what extent pharmacological intervention may be able to counteract stress-induced alterations and promote coping mechanisms to reduce the negative effects of stress exposure. On these premises, this review will discuss selected stress-related mechanisms that may be relevant for the manifestation of schizophrenic symptoms, to evaluate the ability of atypical antipsychotic drugs (AAPDs) to modulate or interfere with such mechanisms. We will highlight the interaction between pharmacological intervention and stress-related changes, providing an overview of the effects of these drugs in counteracting alterations that are critical for the patient’s management. PubMed searches were performed, and the most recent literature was screened: relevant articles were selected if they reported new and useful insights on the topics. However, to provide a deeper knowledge of certain aspects, we also included “milestone” papers of the field.

## 2. Molecular Mechanisms of the Stress Response

Several mediators, including neurotransmitters, hormones, and neuropeptides, are involved in the stress response. Based on the nature and the timing of the stressor, different mediators are released to act on specific brain regions and neural populations, leading to unique downstream effects that are crucial for appropriate adaptive responses [11]. Moreover, the duration of the stressor determines the nature of such a response. Indeed, exposure to acute stress produces the rapid and transient release of neurotransmitters and hormones, which is considered “protective”, to respond to such acute challenge. On the other end, chronic or sustained exposure to stress—lasting one week or more—produces protracted and even permanent changes affecting gene expression, neuronal function, and ultimately brain structures, which may lead to psychopathologic consequences [1,12].

In the following sections, we will focus on the major elements that contribute to stress response.

### 2.1. Monoamines

The exposure to a stressful situation induces a rapid release of monoamines, such as dopamine, noradrenaline, and serotonin, in specific brain regions—primarily within the hippocampus, the amygdala, the prefrontal cortex, and the nucleus accumbens [11]. Stress-induced monoamines release usually occurs within minutes after the onset of the challenge, and it may outlast the duration of the stressor itself. The main pathways activated by monoamines involve G-protein coupled receptors that quickly activate downstream effectors, altering neuronal functionality [11]. Each mediator, within specific brain regions, may contribute to different behavioral changes observed as a consequence of stress exposure. As an example, increased levels of noradrenaline may shift the attention from processing of sensory information to a more general scanning of the overall environment, providing better solutions to endure the challenge [13]; dopamine release may improve risk assessment and decision making [14], while serotonin is crucial to reduce post-stress anxiety [15]. Thus, monoamines cooperate and interact, determining a wide range of behavioral responses that are fundamental to face the initial phase of a challenging situation, and to determine coping strategies.

On the other end, sustained exposure to stressful events may lead to different changes in monoaminergic transmission. Accordingly, preclinical evidence suggests that chronic stress in adult rats increases the levels of serotonin and noradrenaline in several brain regions [16], whereas adolescent chronic stress strongly affects dopaminergic transmission in the medial prefrontal cortex. In more detail, it has been found that stress decreases basal dopamine levels [17], dopamine turnover, and D2 receptors expression [18,19], whereas the expression and binding of the dopamine transporter (DAT) are increased by a chronic challenge [20,21]. These alterations suggest an overall downregulation of dopaminergic activity and function as a consequence of chronic exposure to stress.

### 2.2. Glutamate and γ-aminobutyric Acid

Among other neurotransmitters released after stress exposure, glutamate and γ-aminobutyric acid (GABA) play pivotal roles as they can finely modulate the neuronal excitation-inhibition balance, which is thought to be disrupted in several psychiatric disorders [22,23]. Accordingly, stress exposure in adulthood alters this balance within the prefrontal cortex, leading to the development of emotional disturbances [24,25].

In line with this idea, glutamatergic and GABAergic changes have been observed both in human studies and in animal models of emotional disorders [25]. Stress may differentially impact these systems based on the type and the duration of the stressor itself, as well as on the age and gender of the subjects involved. Accordingly, acute stress increases the excitatory glutamatergic transmission in cortical regions by enhancing glutamate release [26,27]. In addition, it determines delayed changes in postsynaptic glutamate receptors responses, by increasing both *N*-methyl-d-aspartate (NMDA) and α-amino-3-hydroxy-5-methyl-4-isoxazolepropionic acid (AMPA) currents in the prefrontal cortex, facilitating synaptic function and working memory [28]. Contrastingly, chronic stress exposure may determine a loss of glutamate receptors, impairments in glutamate metabolism, and a suppression of glutamate transmission [29], which may contribute to emotional dysregulation [30].

Studies about GABAergic alterations following stress exposure often show contradictory findings, with the majority reporting a decrease in GABAergic functionality, while few data support the idea of an increased inhibition after stress [25]. Moreover, an acute challenge can transiently increase the levels of the GABA synthetic enzymes, GAD67, and GAD65, in brain regions involved in stress response [31]. On the other end, chronic stress impairs GABAergic transmission by decreasing the levels of GAD67 in cortical brain regions [32], as well as by disrupting GABA uptake and release [33]. However, there is evidence of increased GABAergic activity following stress exposure. For instance, chronic stress increases inhibitory synapses onto glutamatergic cells [34] and it induces dendritic hypertrophy in inhibitory interneurons [35]. Moreover, GABAergic interneurons that express parvalbumin (PV) and somatostatin (SST), are extremely vulnerable to the detrimental consequences of stress exposure [36,37]. Indeed, several studies reported that chronic stress decreases *Pv* and *Sst* mRNA levels, as well as the number of neurons expressing these markers [32,36,37,38].

### 2.3. Hypothalamic-Pituitary-Adrenal Axis and Glucocorticoids

The hypothalamic-pituitary-adrenal (HPA) axis is the major neuroendocrine system activated in response to stress. It stimulates the release of neuropeptides, namely corticotrophin-releasing hormone (CRH) and vasopressin (VP), from the paraventricular nucleus of the hypothalamus, which in turn trigger the release of adrenocorticotropic hormone (ACTH) from the pituitary. ACTH binds to its receptors in the zona glomerulosa of the adrenal cortex to stimulate the production and secretion of glucocorticoids (cortisol in humans and corticosterone in rodents) into the bloodstream. Glucocorticoids then act through specific receptors both in the brain and in the periphery, triggering a wide range of metabolic, functional, and neuronal changes that are fundamental for the proper stress response [1,11,39].

Within the brain, basal endogenous glucocorticoids selectively bind the mineralocorticoid receptors (MR), regulating their circadian fluctuations. On the other end, when glucocorticoid levels rise following a stimulus, they bind and activate the glucocorticoid receptors (GR) to modulate the stress response [40]. Indeed, once activated by glucocorticoids, GRs are released from a chaperone complex in the cytosol and migrate into the nucleus, where they regulate the transcription of target genes, by directly interacting with glucocorticoid response elements (GRE) as well as by modulating different transcription factors.

The HPA axis is controlled by a sophisticated negative feedback mechanism, which is necessary to terminate the stress response. In this regard, glucocorticoids act as potent negative regulators of the HPA axis activity: by binding to their receptors in different structures, glucocorticoids extinguish the HPA axis response [39,41]. While the acute activation of the HPA axis is essential for stress adaptation and homeostasis preservation [1], excessive and sustained exposure to stress hormones, such as cortisol and CRH, might determine harmful modifications, leading to a hyperactivity of the axis and predisposing the subject toward the development of psychiatric disorders [39,42]. The hyperactivity of the HPA axis observed in depressed [43] and schizophrenic [44] patients are probably the consequence of a disruption in the negative feedback mechanism, which may originate from a dysfunctional GR activity or expression, a process also known as “glucocorticoid resistance”. Indeed, changes in the expression, nuclear translocation, and transcriptional activity of GRs may concur in altering HPA axis responsiveness to glucocorticoids, which in turn may contribute to HPA axis hyperactivity [45]. Hence, the alterations of the HPA axis—GR activity represent a core element in mental disorders, which concur to the onset of dysfunctional behaviors.

### 2.4. Neuronal Plasticity

Within the brain, different structures and circuits are activated and dynamically coordinate the stress response. Indeed, “neuronal plasticity” is considered as an index for the ability of specific brain structures and circuits to actively modify their connections and function in response to the environment, to adapt through changes aimed to preserve functions, and to store the information of such experience [46]. However, chronic stress exposure could endanger the function of specific circuits or neuronal populations that ultimately lead to pathologic outcomes. Indeed, a bi-directional relationship between stress and brain plasticity has been proposed. Accordingly, mild-to-moderate stressors have a beneficial effect on a wide set of neurobiological endpoints by promoting neural plasticity and the adaptive remodeling of neural circuits [47]. By contrast, exaggerated exposure to stress has detrimental effects on plasticity, leading to sustained changes in brain structures, including neuronal atrophy and synaptic dysfunction that is often associated with psychiatric disorders [1,46,48]. It may be inferred that pharmacological treatments should be able to improve such alterations to be fully effective at the clinical level [49].

Among other genes closely related to neuroplasticity, neurotrophic factors, particularly the neurotrophin brain-derived neurotrophic factor (BDNF), play a pivotal role in regulating different mechanisms relevant for plasticity and synaptic remodeling [50,51,52]. Indeed, while BDNF up-regulation following an acute challenge may represent a pro-survival mechanism to cope with the adverse experience, chronic and protracted exposure to stressors leads to impaired BDNF expression and function, which may contribute to the onset of psychiatric symptoms [1,50].

### 2.5. Inflammation

Neuroinflammation plays an important role in the development and exacerbation of stress-related neuropsychiatric disorders. Indeed, despite cytokines are primarily involved in the response to immunological stimuli, there is a close loop between cytokines and the HPA axis. Indeed, stress itself can induce a pro-inflammatory state within the brain and other systems [53], whereas cytokines can modify the HPA axis responsiveness [54] and are associated with depressed states and cognitive deficits [55,56]. Cytokines production is divided into two major groups based on the type of the secreting T-helper cell: while type 1 cells (Th1) mediate the cellular immune response through the production of cytokines like tumor necrosis factor α (TNF-α) and interleukin 2 (IL-2); type 2 cells induce immune reactions involving antibodies and determine the production of IL-6 and IL-10 [53,57]. It has been proposed that stress may alter the balance of type 1 and type 2 responses: stress-induced neuroendocrine mediators may selectively suppress Th1 responses [58]. Therefore, it is believed that stress mediators may disrupt the anti-/pro-inflammatory balance, resulting in increased pro-inflammatory responses. Nevertheless, depending on its nature, stress might also activate anti-inflammatory pathways as a neuroprotective mechanism [53].

## 3. Stress Exposure and Schizophrenia

Schizophrenia is a chronic psychiatric disorder that affects approximately 20 million people worldwide. It is a complex and severe disease characterized by three major symptomatologic domains. Positive symptoms are clearly identified as psychosis, which encompasses hallucinations, thought disorganization, delusions, and abnormal motor behaviors. Negative symptoms are associated with disruptions of emotions and behaviors and they include anhedonia, avolition, and emotional flattening. Cognitive dysfunction is associated with different alterations, including poverty of speech, attention and memory impairments, and poor executive functioning [59]. Along with this tangled variety of symptoms, schizophrenia is characterized by a complex etiopathogenesis. Indeed, it is well established that genetic and environmental factors concur in the onset of the illness. Within this context, exposure to stress may contribute to the development of schizophrenia in predisposed individuals and may also represent an important risk factor for relapse.

### 3.1. Clinical and Epidemiological Evidence

It is known that specific life stages are characterized by enhanced sensitivity and vulnerability to stress. During pregnancy, for instance, the fetal brain undergoes fast anatomical and functional development, and maternal inputs are fundamental to ensure its correct maturation. Along with the nutritional and physical state, the emotional well-being of the mother has a deep impact on the fetus’s health. Indeed, evidence from many prospective studies has shown that children born from anxious, depressed, or stressed mothers during pregnancy, are more likely to experience adverse neurodevelopmental fallouts, increasing the risk for psychopathologies later in life [60,61]. Accordingly, consistent evidence has demonstrated that the experience of serious stressful life events during pregnancy has been associated with the development of psychotic symptoms later in life and increased incidence of schizophrenia in the offspring [62,63]. It is assumed that these consequences may originate from sustained exposure to maternal glucocorticoids that interfere with the development and maturation of the fetal brain circuits [64]. Moreover, excessive glucocorticoids may induce an overall “sensitization” of the fetal brain to their action, potentially determining an exaggerated response to subsequent exposure to stress.

Childhood represents another critical and fragile developmental period. Childhood trauma, including physical, psychological, or sexual abuse, as well as physical and emotional neglect, is a severe form of stress that sharply increases the risk of developing psychosis and schizophrenia later in life [65,66]. In this regard, physical abuse and emotional neglect experienced during childhood have been widely reported by several schizophrenic patients [67], and these experiences are known to negatively modulate brain maturation, potentially affecting cognition and memory [68,69,70], which are impaired in schizophrenia.

Moreover, adolescence is a developmental stage of elevated risk for the outbreak of mental disorders. Indeed, while many patients manifest premorbid alterations already during adolescence, such as social withdrawal, thought abnormalities, cognitive deficits, and hormonal imbalance [71,72], the outbreak of schizophrenia symptomatology usually occurs at the transition between adolescence and adulthood [73]. In this regard, stress may increase such vulnerability or precipitate the development of psychotic symptoms [74].

Further corroborating the relevance of stress in the disorder, schizophrenic patients show elevated cortisol and ACTH levels [44,75]. In addition, postmortem data suggest a reduction in glucocorticoid receptors levels in patients with schizophrenia [76], and such dysregulation may in turn lead to the increased secretion of glucocorticoids [77]. The HPA axis abnormalities can contribute to schizophrenia symptomatology as well as to structural changes observed in the brain of schizophrenic patients. Indeed, a close-loop may exist between glucocorticoid alterations, HPA axis functionality, reduced hippocampal volume, and cognitive dysfunction [44,78].

### 3.2. Preclinical Studies and Animal Models

Considering the role of stress in enhancing the risk for schizophrenia-related disorders, several animal models have been used to characterize such a link and to identify the potential neurobiological underpinnings.

Several studies have shown that prenatal stress (PNS) exposure promotes the development of behavioral alterations that resemble schizophrenic symptoms. For instance, PNS exposure determines cognitive deficits [79,80,81], the development of depressive-like states [82,83], and anhedonic-like behaviors [84]. Moreover, we have demonstrated that PNS exposure determines a drastic reduction of social behaviors in periadolescent rats [5], mimicking the social withdrawal that may be experienced by adolescent patients.

It is known that PNS induces long-lasting alterations in plasticity-related systems, that may account for some of the observed behavioral alterations. Indeed, we have previously reported that this manipulation drastically reduces the expression of neurotrophic factors, such as BDNF and fibroblast growth factor 2 (FGF-2) [5,85,86,87]. Moreover, it has been demonstrated that exposure of pregnant dams to restraint stress during the last week of gestation alters the corticolimbic dopaminergic and corticostriatal glutamatergic systems of the offspring, with an effect that persists into adulthood [88]. Accordingly, PNS modulates the expression of key glutamatergic markers, such as the NMDA receptor subunits NR-2A and NR-2B, and the scaffolding postsynaptic density protein 95 (PSD-95) [89]. Furthermore, PNS exposure leads to a reduction of metabotropic glutamate receptor 2, paired with overexpression of serotoninergic receptor 2A in the frontal cortex [90]. Long-lasting changes in the glutamatergic transmission may increase the susceptibility to a subsequent challenge. Indeed, we have shown that PNS can alter glutamatergic responsiveness to an acute challenge in adulthood, preventing its activation via phosphorylation of NMDA receptor subunits NR-1 and NR-2B [91]. Together with glutamatergic alterations, PNS disrupts GABAergic transmission, resulting in an overall reduced activity [92]. Accordingly, PNS exposure alters the embryonic neurogenesis of GABAergic progenitor cells [93] and their migration to the cortex [94], potentially increasing the vulnerability toward mental disorders. Along with changes in neurotransmission, prenatal stress in rodents has been associated with disruptions of the HPA axis functionality and reduction of GR and MR expression in the hippocampus [95,96]. Moreover, PNS exposure increases circulating corticosterone levels [97], as well as SGK1 expression within the rat hippocampus [98].

The adolescent period in rats generally ranges between postnatal days (PND) 28–42 and it is characterized by several behavioral and neuroanatomical changes [6,99], which may lead to enhanced vulnerability to the detrimental effects of stress. Accordingly, Gomes and colleagues reported that strong stressors occurring during adolescence may determine dopaminergic alterations in adulthood, as indicated by enhanced locomotor responses to amphetamine, as well as cognitive deficits [100]. These changes are associated with increased spontaneous activity of dopaminergic neurons, suggesting that early-life adversities may profoundly affect brain function [100]. Along this line of reasoning, social isolation in the peripubertal period (adolescence) produces a wide range of long-lasting alterations that may resemble core features of schizophrenia [101,102]. Indeed, several studies have reported that socially isolated rats develop an increased reactivity to a novel environment with locomotor hyperactivity, which resembles positive traits of schizophrenia (for a detailed review, see [102]). Moreover, exposure of rodents to an early-life social challenge may increase the sensitivity to the effects of psychostimulant drugs, among which amphetamine, determining an exacerbation of the hyperactivity state [103,104]. Furthermore, social isolation may induce a persistent anhedonic-like phenotype [105,106], with a disruption of sensorimotor gating, a measure of the ability to integrate cognitive and sensory inputs, as well as a deficit in novel object discrimination paradigm [107,108], reflecting attention and memory dysfunctions. The behavioral alterations following stress in adolescence are associated with different neurochemical and molecular changes. For instance, isolation rearing may decrease dopamine turnover in the medial prefrontal cortex, which resembles the hypo-frontality seen in schizophrenia [101]. Challenges occurring during adolescence may also induce disruptions in neuroplasticity-related markers. Accordingly, we have recently demonstrated that adolescent social isolation in male rats determines a selective reduction of *Bdnf* mRNA levels within the prefrontal cortex, but not the hippocampus [105]. Moreover, isolation rearing can impair the HPA axis functionality, highlighting the strong impact on stress-related responses. For instance, Boero and colleagues reported that social isolation decreased both total corticosterone and its carrier levels, whereas it increased the expression of GR paired by a reduction of MR levels [109]. In addition, we have found that *Nr3c1* (the gene encoding for GR) mRNA levels were increased in the prefrontal cortex of isolated male rats, whereas they were reduced in the ventral hippocampus of female animals. We also observed increased expression of the chaperone *Fkbp5* in the ventral hippocampus of male isolated rats [105].

Stress is also an important player in adulthood as it may exacerbate preexisting symptoms or lead to relapse in vulnerable individuals. Thus, studying the effects of acute and chronic stress occurring during adulthood is crucial to identify those alterations that may be modulated by pharmacological intervention. For instance, chronic mild stress (CMS) exposure induces several behavioral alterations, including anhedonia and cognitive impairment [110,111,112,113]. These behavioral alterations are associated with a wide range of molecular changes affecting specific systems that are altered in schizophrenia as well as in other mental disorders. For instance, we have shown that CMS exposure reduces the expression of neuroplastic proteins [114] and alters synaptic mechanisms involved in the glutamatergic function and local protein synthesis [111]. Moreover, exposure to CMS is associated with alterations in the glucocorticoid signaling [113], enhanced expression of inflammatory markers as well as an imbalance of redox mechanisms [38,110], all of which have been associated with schizophrenia.

Altogether, preclinical studies demonstrate that exposure to stress at different stages of life produces complex behavioral outcomes resembling specific psychopathologic domains, which are associated with several molecular alterations in different brain structures. Since these changes may sustain the functional impairment and the increased vulnerability for psychopathology, they represent an important target for pharmacological intervention.

## 4. Mechanism of Action of Atypical Antipsychotics

Antipsychotic drugs are the standard therapy for schizophrenia. They are commonly classified in typical and atypical antipsychotics (also known as first (FGA) and second-generation (SGA), respectively) based on the notion that atypicals (AAPDs) show reduced extrapyramidal side effects (EPS), and they may be more effective on the cognitive and social dysfunction associated with schizophrenia [115,116].

The group of AAPDs comprises a large number of molecules that, while sharing some mechanisms, must be considered unique on the bases of their receptor and synaptic profiles, but also considering their ability in modulating intracellular mechanisms downstream from receptors. Indeed, although dopamine D_2_ receptors represent a common target for all antipsychotics, the activity of AAPDs on these receptors is significantly less—when compared to old drugs—which may normalize dopaminergic transmission without producing significant side effects, such as EPS and hyperprolactinemia. For example, molecules like clozapine and quetiapine show a fast dissociation rate from such receptors, while other drugs, including aripiprazole, brexpiprazole, and cariprazine, are partial agonists at dopamine D_2_ receptors [116]. Along with dopaminergic modulation, AAPDs are also antagonists at serotoninergic 5-HT_2A_ receptors [117], a mechanism that may modulate dopamine release in different brain regions, leading to reduced incidence of motor side effects, while providing a certain extent of improvement in negative and cognitive symptoms of schizophrenia [117]. Moreover, as reviewed by Aringhieri and colleagues, AAPDs may directly or indirectly modulate other subtypes of dopaminergic and serotoninergic receptors, as well as other receptor subtypes [116]. These different receptor mechanisms may also regulate neurotransmitters release within specific brain regions relevant to schizophrenia symptomatology. As an example, different AAPDs can increase the release of dopamine within the prefrontal cortex, while decreasing dopaminergic transmission within the nucleus accumbens [117]. Moreover, AAPDs can potentiate the impaired glutamatergic system [118,119], while modulating striatopallidal GABAergic neurons [117]. It is believed that the multifaceted mechanism of action of AAPDs may also involve the modulation of neuroplasticity-related markers as well as neurogenesis [116]. However, such multifaceted mechanisms of action may also be associated with the occurrence of a wide range of side effects. Indeed, AAPDs might induce metabolic adverse effects—weight gain, lipidic and glycemic imbalance—as well as electrolyte imbalance and cardiovascular abnormalities [120,121].

It is feasible to hypothesize that, while specific receptor mechanisms regulate selected functional domains by interfering with key neurotransmitter systems, synaptic mechanisms may also regulate the activity of given circuits and initiate adaptive changes that will eventually contribute to restoring altered functional domains. On these bases, it is of interest to establish if and how AAPDs may modulate stress response by acting on the systems that are directly affected by the exposure to challenging events.

## 5. Atypical Antipsychotic Drugs and Stress-Related Mechanisms

### 5.1. Preclinical Studies

Given the crucial role of stress in the development of schizophrenia, the modulation of stress-related mechanisms may provide additional therapeutic benefit to pharmacological intervention. Accordingly, several preclinical studies have investigated the modulatory effects of AAPDs on stress-induced behavioral and molecular alterations in different experimental settings. The major stress-related changes as well as the modulatory activity of AAPDs are summarized in Table 1.

As already mentioned, the exposure to an acute stressor can activate key brain areas that may be involved in psychotic manifestations. To this point, repeated administrations of lurasidone in adult rats determine an enhanced increase in the expression of *Bdnf* expression following acute swim stress. This evidence suggests that chronic lurasidone treatment may facilitate coping mechanism when facing a challenge [122]. On a different note, we have recently reported that the chronic treatment with the AAPD blonanserin can prevent the strong increase in immediate-early genes expression that follows acute swim stress, suggesting a potential protective mechanism in counteracting the exaggerated brain activation following an adverse experience [123].

Different AAPDs appear to exert positive effects on the behavioral and molecular alterations produced by different paradigms of chronic stress, which produce a wide range of functional alterations that resemble specific pathologic domains found in several psychiatric conditions.

Considering the role of early-life stress in the development of schizophrenia-related conditions, different studies have investigated the ability of AAPDs in counteracting and/or preventing the changes produced by such adverse experiences. For instance, sub-chronic treatment of adult rats with clozapine, but not haloperidol, can ameliorate prenatal stress-induced behavioral alterations, including locomotor hyperactivity, social interaction, pre-pulse inhibition, and fear conditioning deficits [124,125]. Along with behavioral alterations, PNS may disrupt the epigenetic mechanism by enhancing DNA methylation through persistently increased expression of DNA methyltransferases [124]. With this regard, clozapine—but not haloperidol—is also able to prevent the PNS-induced increase of DNA-methyltransferase 1 and ten-eleven methylcytosine dioxygenase 1 levels, resulting in reduced DNA methylation at the promoters of *Bdnf* and *Gad1*, two key genes associated with the development of psychiatric disorders [125]. Moreover, clozapine also normalizes the reduced expression of *Bdnf*, *Gad1*, and *Reelin* mRNAs and proteins following PNS exposure [125]. In line with these findings, we have also shown that PNS exposure has a long-lasting negative effect on neuroplasticity, with a significant reduction of *Bdnf* expression, an effect that becomes evident in adulthood. To this point, we have demonstrated that early pharmacological intervention (i.e., during adolescence) with the AAPD lurasidone, is able to prevent the PNS-induced reduction of *Bdnf* in adulthood [87].

The effects of antipsychotic drugs have also been investigated following exposure to stress during early postnatal life. As an example, treatment with olanzapine during adolescence tends to counteract some of the alterations produced by early maternal deprivation (MD), including the reduction in body weight, corticosterone responses, and the reduced expression of the cannabinoid CB1 receptor [126]. In addition, the AAPD blonanserin can ameliorate social deficits, assessed in the social interaction test, produced following a combination of prenatal and adolescent stressors [127]. Treatments with AAPDs are also able to normalize cognitive performances in adolescent rats: for instance, risperidone can revert some of the cognitive deficits that follow adolescent social isolation [128], while quetiapine and olanzapine normalize the deficits observed in pre-pulse inhibition [129]. On a different note, clozapine administration in adulthood can normalize the pathologic phenotype of animals exposed to post-weaning social isolation and reverse the imbalance of redox mechanisms by increasing the activity of the antioxidant enzyme superoxide dismutase (SOD) and regulating the glutathione oxidation balance in the frontal cortex and striatum [130]. Isolation rearing from weaning also determines a rise in pro-inflammatory cytokines, linked to metabolic disturbances, increased locomotor activity, and impaired PPI. Chronic quetiapine administration at adolescence can ameliorate these dysfunctions as well as restore normal levels of pro-inflammatory cytokines [131].

As already mentioned, exposure to stress in adulthood may be relevant for psychiatric disorders, including schizophrenia, since it may precipitate the pathologic condition and contribute to relapse. With this respect, different preclinical studies have demonstrated that pharmacological intervention with AAPDs is effective in ameliorating several alterations produced by adult chronic stress. It has been reported that the treatment of adult rats with olanzapine for 3 weeks can reverse depression- and anxiety-like behaviors observed following social isolation, as well as to prevent the stress-induced increase in c-Fos expression in different brain regions [132]. On the other end, chronic treatment with clozapine can revert the social isolation-induced deficits in the number of parvalbumin-positive interneurons within specific hippocampal sub-fields, thus exerting a protective effect from the deleterious effects of stress [133].

Exposure of adult rats to chronic mild stress (CMS) can induce anhedonic-like behaviors that are successfully normalized through the chronic administration of several AAPDs, such as lurasidone, cariprazine, and olanzapine [110,114,134,135], suggesting their potential in ameliorating emotional disturbance, which represents a major burden in different psychiatric disorders. Moreover, we have recently demonstrated that chronic lurasidone administration is also able to revert the cognitive deficit induced by CMS exposure [113] and a similar effect was also detected with olanzapine that was able to improve the impairments in visual and spatial performances [136]. Furthermore, chronic olanzapine administration ameliorates other behavioral alterations following CMS exposure, including deficits in coat state and grooming behavior in the splash test, while decreasing the attack frequency in the resident intruder test and the immobility time in the tail suspension test [137].

Despite the pivotal role of the HPA axis in the stress response, limited preclinical data are available on the effects of AAPDs in improving stress-induced changes at this level. Although the HPA axis does not represent a direct target of AAPDs, several drugs may modulate key brain regions linked to the stress response [117,138]. For instance, we have recently demonstrated that chronic administration of lurasidone can prevent the increase of GR membrane levels that followed CMS exposure, as well as to restore the transcription of GR-responsive genes [113].

As already mentioned, since neuroplasticity is a fundamental process involved in the stress response as well as in the development of psychiatric disorders, it may represent a potential target for therapeutic intervention. Accordingly, AAPDs modulate several plasticity-related markers that are altered by different stress paradigms. As an example, chronic treatment with lurasidone can normalize the expression levels of *Bdnf* in the prefrontal cortex of animals exposed to CMS, an effect that correlates with the ability of lurasidone in normalizing the anhedonic phenotype [114]. Similarly, chronic olanzapine administration in adult mice effectively counteracts the CMS-induced decrease of *Bdnf* and *Creb* levels [139].

Imbalance in excitatory/inhibitory mechanisms may result from stress exposure and it is another key alteration observed in schizophrenic patients [140]. In this regard, GABAergic transmission has been proposed as a potential target of antidepressants and antipsychotic drugs [141,142]. For instance, chronic treatment with lurasidone in rats can normalize the CMS-induced reduction of *Parvalbumin* gene expression within the dorsal hippocampus, an effect that may depend on its ability in regulating the oxidative balance in this brain region [38]. In more detail, chronic administration of lurasidone for 5 weeks can counteract the CMS-induced increase of the pro-oxidant enzyme NADPH oxidase 2 (NOX2) levels as well as the reduced levels of antioxidant mediator nuclear factor erythroid 2-related factor 2 (NRF2) in the dorsal hippocampus [38]. These results are in line with another report showing that chronic administration of paliperidone regulates the redox machinery by counteracting the reduction of NRF2 in the prefrontal cortex of rats exposed to chronic restraint stress [143].

**Table 1 pharmaceuticals-13-00322-t001:** Behavioral and molecular changes in stress-based animal models of psychiatric disorders and their modulations following atypical antipsychotic drug (AAPD) treatment.

Stress Model	Protocol	Major Changes	Effects of AAPD Treatment
Prenatal stress (PNS)	Pregnant dams are exposed to repeated immobility sessions in a plastic tube (usually during the last week of gestation)	Cognitive deficits [81,91]; locomotor hyperactivity; social interaction, pre-pulse inhibition and fear conditioning deficits [124,125], anhedonia [84] HPA axis dysfunctions [95] Reduction of *Bdnf* [5,87,125] Epigenetic changes [124,125]	Amelioration of some schizophrenic-like behaviors [124,125] Restoration of *Bdnf* levels [87,125] Limited DNA methylation of promoters of key genes implied in schizophrenia [125]
Post-weaning social isolation (PWSI)	Animals are reared in isolation during the peripubertal phase. They are kept in the same room but are prevented from physical contacts with conspecifics	Cognitive and PPI deficits [128,129], locomotor hyperactivity [102], anhedonia [105] HPA axis dysfunctions [109] Impaired expression of neuroplasticity markers [105,132] Impaired oxidative balance [130]	Amelioration of cognitive and PPI deficits [128,129] Restoration of key oxidative markers [130] Restoration of *c-Fos* [132] and *Parvalbumin* [133] levels
Chronic mild stress (CMS)	Animals are exposed to different mild stressors presented in unpredictable sequences	Anhedonia [110,134], cognitive deficits [113,136] Alteration in glucocorticoid signaling [113] Enhanced expression of inflammatory markers, imbalance of redox mechanisms [38,110] Impaired expression of neuroplasticity markers [114]	Amelioration of anhedonia [110,134,135] and of cognitive deficits [113,136] Restoration of glucocorticoid signaling [113] Modulation of inflammatory [110,137] and redox markers [38] Normalization of *Bdnf* expression [114]

Interestingly, chronic treatment with lurasidone also improves CMS-induced neuroinflammation, by restoring IL-1ß levels [110]. Similarly, chronic administration of olanzapine for 5 weeks prevents the increase of IL-6 levels that follows CMS exposure in adult male mice [137]. Last, CMS may alter several metabolic pathways and, accordingly, Cai and colleagues investigated the effects of AAPDs (aripiprazole, clozapine, and risperidone) on stress-induced metabolic alterations in the prefrontal cortex and hippocampus [138]. They have shown that treatment with AAPDs can improve the stress-induced deficits, by increasing the levels of key metabolic markers, like creatine, progesterone, and phosphatidylethanolamines [138].

Other chronic stress paradigms have been used to investigate the effects of AAPDs in adult animals. As an example, the chronic administration of ziprasidone can prevent the increase of corticotropin-releasing factor (CRF) in the hypothalamic paraventricular nucleus (PVN) of rats that follows chronic immobilization stress [144]. Furthermore, another study reported that the administration of risperidone at ultra-low doses was able to revert the stress-induced rise in circulating corticosterone levels [145], highlighting the potential anti-stress properties of this drug. Moreover, immobilization stress decreases *Bdnf* mRNA levels in the hippocampus and neocortex, and chronic ziprasidone treatment is effective in attenuating this deficit [146]. Similarly, quetiapine administration can attenuate the reduction of BDNF hippocampal protein levels [147], as well as to reverse the stress-induced suppression of hippocampal neurogenesis [148].

Taken together, preclinical studies provide evidence that AAPDs can ameliorate—or even prevent—behavioral, functional, and molecular changes set in motion by stress exposure at different stages of life.

### 5.2. Clinical Evidence

Despite the consistent preclinical evidence for AAPDs in exerting a positive effect on the stress response, more limited information exists at the clinical level. However, as stress is a key environmental factor that may be involved in the onset of schizophrenia, the characterization of stress-related mechanisms in patients represents an important strategy for intervention. Table 2 provides an overview of the main stress-related changes observed in schizophrenic patients and the modulatory activity by AAPDs.

For instance, it has been reported that schizophrenic patients often show maladaptive responses to stress, and alterations in the HPA axis mechanisms may play a pivotal role [8,9]. Several studies have indeed reported elevated cortisol levels in patients, as compared to healthy controls [9,149]. As hypercortisolemia has been associated with negative symptoms of schizophrenia [150], the improved efficacy of atypical versus typical antipsychotics on this domain may lie in the ability of AAPDs to normalize cortisol levels. Indeed, some studies have demonstrated that treatment with AAPDs, as clozapine and olanzapine, reduces cortisol levels as compared to treatment with typical drugs [151,152,153], even if not all studies confirmed such effect [154]. Supporting this evidence, the administration of quetiapine and olanzapine in healthy subjects produces a reduction of plasma ACTH and cortisol levels, as compared to placebo, an effect that was not observed with the classical antipsychotic drug haloperidol [155]. The modulatory activity of AAPDs on the HPA axis suggests a potential effect on the stress response, which may produce clinical benefits in ameliorating depressive-like states, negative symptoms, and cognitive deterioration. Indeed, as the HPA axis plays a key role in the onset of the different pathologic domains of schizophrenia, it is worth pointing out that different AAPDs can ameliorate comorbid depressive symptoms, as well as anxiety-related manifestations [156,157,158,159]. In addition, growing evidence suggests that the immune-endocrine crosstalk may also be defective in schizophrenic patients. Accordingly, a 12-week treatment with risperidone decreased the elevated cortisol levels and improved negative symptoms, while producing effects similar to haloperidol on positive symptoms and two cytokines, IL-2 and IL-6 [160]. In another study, a group of schizophrenic subjects had elevated baseline levels of IL-1β and TNF-α, with a reduction of BDNF. An 11-week treatment with risperidone decreased the high levels of plasma IL-1β while elevating those of BDNF, with a concomitant reduction of the Positive and Negative Syndrome Scale (PANSS) scoring. However, the levels of the pro-inflammatory cytokine TNF-α plasma were increased following pharmacological treatment, suggesting potential toxicity after long-term risperidone administration [161].

As already said, brain plasticity may represent a crucial target for the effectiveness of schizophrenia treatment. Indeed, AAPDs administration has been linked to positive effects on brain volume as compared to first-generation antipsychotic drugs [162,163,164]. For instance, diffusion tensor imaging showed that schizophrenic patients have altered the organization of white matter as well as reduced nodal and connectivity patterns. Risperidone administration attenuated the alterations of nodal and connectivity to a major extent, as compared to clozapine, and was linked to improved cognitive performances [165]. Furthermore, schizophrenic patients often show reduced volumes of grey and white matter in the caudate nucleus. It has been shown that treatment with olanzapine increased the volumes of both grey and white matter while ameliorating both positive and negative symptoms [166]. Interestingly, chronic treatment with clozapine in schizophrenic patients was associated with increased serum BDNF levels, while typical antipsychotic drugs did not produce this effect [167]. A strong increase in peripheral BDNF levels was also observed following chronic risperidone administration [161]. Furthermore, a recent study compared the effects of 6-week monotherapy treatment with lurasidone or olanzapine, showing that both drugs were able to increase serum BDNF levels. Moreover, they reported an increase in Nerve Growth Factor (NGF) and Neurotrophin 3 levels. Such changes were associated with a significant clinical improvement of PANSS scoring as well as of social and occupational functionality [168]. In addition, a recent pharmacogenomic study was able to demonstrate that synapse-related genes are strong predictors for the clinical response to the AAPD lurasidone [169]. Since synaptic mechanisms are impaired by stress, this finding provides “translational” support to the idea that lurasidone may exert, at least in part, its therapeutic effects by counteracting stress-induced alterations at the synaptic level [114]. Altogether, this evidence suggests that AAPDs possess multiple neuroprotective properties, which may underlie their wide spectrum activity.

Last, since stress may precipitate psychotic symptoms, leading to relapse [8], one important consequence for the ability of AAPDs in modulating stress responsiveness is the reduced relapse risk in the long-term. Indeed, treatment with AAPDs appears to decrease the risk for relapse and to increase resilience [170], which may lead to beneficial consequences on patients’ quality of life [171].

**Table 2 pharmaceuticals-13-00322-t002:** Main stress-induced alterations observed in schizophrenic patients and their modulations following AAPD treatment.

Domains	Stress-Induced Alterations	Main Effects of AAPD Treatment	Molecules
HPA axis	Increased cortisol levels [9,149]	Reduction of cortisol levels [151,152,153,160]	Clozapine, olanzapine, risperidone
Neuroplasticity	Reduction of BDNF, reduced hippocampal volume [78,161]	Normalization of BDNF levels [161,167], NGF and Neurotrophin 3 [168] Positive effects on brain volume [162,163,164]	Risperidone, clozapine, lurasidone, olanzapine
Inflammation	Increase baseline levels of IL-1β and TNF-α [161]	Normalization of IL-1β and increased TNF-α serum levels [161]	Risperidone

## 6. Concluding Remarks

Schizophrenia is a complex mental disease whose treatment is still arduous, and that is characterized by a significant percentage of patients who do not show adequate response to pharmacological intervention [172].

A comprehensive understanding of the mechanisms relevant for drug effectiveness is therefore critical to advance our knowledge and improve the impact of the pharmacological intervention of schizophrenia. While receptor mechanisms play a crucial role in correcting the unbalance in neurotransmitter function, we propose that the investigation of antipsychotic drugs on etiological mechanisms is crucial to delineate a profile of the systems that may contribute to clinical effectiveness. Considering the impact of stress in everyday life, antipsychotic drugs must be able to correct, improve, and possibly prevent, the wide arrays of functional and molecular abnormalities that originate as a consequence of stress exposure at different stages of life (Figure 1).

Along this line of reasoning, preclinical findings suggest that AAPDs may not only normalize neurotransmitter dysfunction in schizophrenia but, through the modulation of different pathways and circuits, they can also improve several stress-related mechanisms that are associated and contribute to the functional alterations observed in schizophrenic patients. Accordingly, clinical observations support the idea that AAPDs may possess anti-stress properties, which may be a key mechanism underlying their wide efficacy. Nevertheless, further studies are needed to fully elucidate this aspect and to identify the systems that are crucial for the therapeutic activity of AAPDs, to develop novel and more effective agents for the cure of schizophrenia.

## Figures and Tables

**Figure 1 pharmaceuticals-13-00322-f001:**
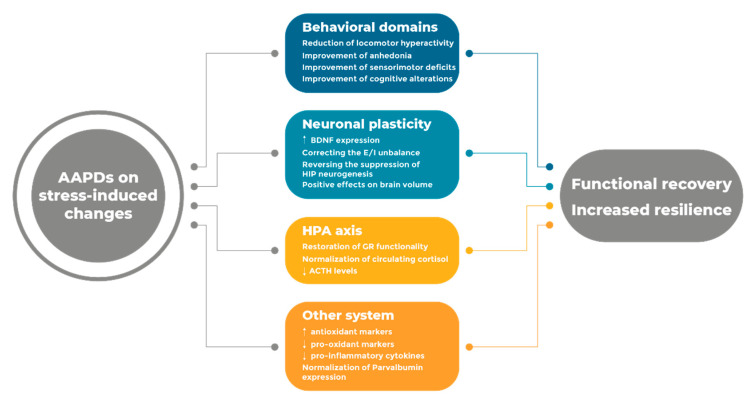
Summary of the major effects produced by AAPDs on stress-induced changes.

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
