# Peer review of "Anti-stress Properties of Atypical Antipsychotics"

_pharmaceuticals, 2020, doi:10.3390/ph13100322_

Round 1
Reviewer 1 Report
Well-written manuscript and timely information. Congratulations on an excellent review.
Author Response
Well-written manuscript and timely information. Congratulations on an excellent review.
We thank the reviewer for the appreciation. We are glad he/she enjoyed reading our work.
Reviewer 2 Report
This review paper attempted to comprehensively describe, with much evidence mainly on the findings from animal models, that the properties of atypical antipsychotic drugs (AAPDs) could involve the ability to modify stress-related molecular alterations which might be the basis of the pathogenesis of mental diseases especially schizophrenia and mood disorder. This is well written and illustrated paper. The discussions were clearly presented and the conclusions were hardly controversial. This paper is fine as far as it goes, but I would like to have several comments.
General comments
- In this paper, the authors listed up inflammatory cytokines as stress-related molecules, but the relationship between stress and inflammation in the brain was not fully explained. It would be easier to understand if the authors would describe it in Section 2.
- The description in this paper seems to be too much weighted for evidence based on animal models compared to human evidence. Thus, in 4.2. clinical evidence, it would be better if the authors would also describe the evidences based on human brain imaging studies and postmortem brain researches.
- In 4.1, the authors describe the potential for epigenetic changes, including DNA methylation, as an effect of prenatal stress and the potential for prevention of this by clozapine. However, it does not fully describe the mechanism by which stress causes epigenetic changes. Maternal starvation during the fetal period and starvation during infant may increase the risk of schizophrenia, and epigenetic changes have been considered as one of the underlying mechanisms. Thus, it would be more interesting to describe it in 2.5.
Minor comments
- The word "effortlessly" on line 169 feels a little strange. The authors should replace it with another word such as apparently, clearly, obviously, manifestly.
Author Response
This review paper attempted to comprehensively describe, with much evidence mainly on the findings from animal models, that the properties of atypical antipsychotic drugs (AAPDs) could involve the ability to modify stress-related molecular alterations which might be the basis of the pathogenesis of mental diseases especially schizophrenia and mood disorder. This is well written and illustrated paper. The discussions were clearly presented and the conclusions were hardly controversial. This paper is fine as far as it goes, but I would like to have several comments.
General comments
In this paper, the authors listed up inflammatory cytokines as stress-related molecules, but the relationship between stress and inflammation in the brain was not fully explained. It would be easier to understand if the authors would describe it in Section 2.
We thank the reviewer for raising this issue. We have therefore further described the relationship between stress and inflammation. Please see a brief description in paragraph 2.5, lines 169-183.
The description in this paper seems to be too much weighted for evidence based on animal models compared to human evidence. Thus, in 4.2. clinical evidence, it would be better if the authors would also describe the evidences based on human brain imaging studies and postmortem brain researches.
We are aware that the review is more focused on preclinical evidences due to the ‘paucity’ of clinical data on this topic. However, in order to accomplish the reviewer’s suggestion, we have described data from brain imaging studies in paragraph 4.2 (lines 501-508).
In 4.1, the authors describe the potential for epigenetic changes, including DNA methylation, as an effect of prenatal stress and the potential for prevention of this by clozapine. However, it does not fully describe the mechanism by which stress causes epigenetic changes. Maternal starvation during the fetal period and starvation during infant may increase the risk of schizophrenia, and epigenetic changes have been considered as one of the underlying mechanisms. Thus, it would be more interesting to describe it in 2.5.
We agree with the reviewer in pointing to epigenetic mechanisms as key elements for stress response. However, we did not quote them in section 2, since we focused on the main systems that are activated and modulated in response to stress rather than on the molecular mechanisms that may lead to such changes.
Minor comments
The word "effortlessly" on line 169 feels a little strange. The authors should replace it with another word such as apparently, clearly, obviously, manifestly.
We replaced the word ‘effortlessly’ with ‘clearly’ (line 188).
Reviewer 3 Report
In the present review, the Authors aimed to evaluate existing clinical and preclinical evidences that may support the ability of atypical antipsychotic drugs to modulate stress-related alterations.
Overall, I found the review very important, timely, well written and scientifically sound: enjoyed reading it! I have only some minor suggestions aimed to improve the high quality of the paper and these are outlined below:
1) I suggest to add how literature was chosen and relevant articles selected.
2) As well, I believe that a table regarding clinical evidences in humans concerning "Atypical antipsychotic drugs and stress-related mechanisms" with relevant studies would be useful to the reader.
3) In the Paragraph "Mechanism of action of atypical antipsychotics", I believe that a a table with the mechanisms of action of newer AAPs would be useful. Moreover, some adverse effects of AAPs may be related to their mechanism of action. This should be added with appropriate references (see Orsolini et al. Expert Opin Drug Saf. 2020 Aug;19(8):981-998 and Expert Opin Drug Saf. 2016 Oct;15(10):1329-47.)
Author Response
In the present review, the Authors aimed to evaluate existing clinical and preclinical evidences that may support the ability of atypical antipsychotic drugs to modulate stress-related alterations.
Overall, I found the review very important, timely, well written and scientifically sound: enjoyed reading it! I have only some minor suggestions aimed to improve the high quality of the paper and these are outlined below:
1) I suggest to add how literature was chosen and relevant articles selected.
We thank the reviewer for the suggestion. We used PubMed to search literature and we thoroughly reviewed the most recent articles. We selected the papers that provided a good insight on the topic. We added this description in the introduction paragraph (lines 52-55).
2) As well, I believe that a table regarding clinical evidences in humans concerning "Atypical antipsychotic drugs and stress-related mechanisms" with relevant studies would be useful to the reader.
We agree with the reviewer that it would be useful to include a table covering the modulation of stress mechanisms in clinical studies. Accordingly, we added a summary table at the end of paragraph 4.2.
3) In the Paragraph "Mechanism of action of atypical antipsychotics", I believe that a table with the mechanisms of action of newer AAPs would be useful. Moreover, some adverse effects of AAPs may be related to their mechanism of action. This should be added with appropriate references (see Orsolini et al. Expert Opin Drug Saf. 2020 Aug;19(8):981-998 and Expert Opin Drug Saf. 2016 Oct;15(10):1329-47.)
We thank the reviewer for raising these issues. A short description of the side effects of AAPDs can be found in paragraph 3, lines 336-339. We did not include a table on receptor mechanisms of AAPDs since we believe that it may not properly reflect the complex receptor profiles that characterize each drug.